# Defect Detection in CFRP Concrete Reinforcement Using the Microwave Infrared Thermography (MIRT) Method—A Numerical Modeling and Experimental Approach

**Sam Ang Keo** [1,*] (ID)**, Barbara Szymanik** [2,*] (ID)**, Claire Le Roy** [3]**, Franck Brachelet** [4] **and Didier Defer** [4] (ID)

1 Cerema, Research Team ENDSUM, 23 Amiral Chauvin Avenue, 49130 Les Ponts-de-Cé, France
2 Center for Electromagnetic Fields Engineering and High-Frequency Techniques,
  Faculty of Electrical Engineering, West Pomeranian University of Technology, 70-310 Szczecin, Poland
3 Independent Researcher, Sikorski St. 37, 70-313 Szczecin, Poland; claire29370@yahoo.fr
4 Univ. Artois, IMT Nord Europe, Junia, Univ. Lille, ULR 4515, Laboratoire de Génie Civil et
  géo-Environnement (LGCgE), 62400 Béthune, France; franck.brachelet@univ-artois.fr (F.B.);
  didier.defer@univ-artois.fr (D.D.)
* Correspondence: keo_samang@yahoo.com (S.A.K.); szymanik@zut.edu.pl (B.S.)

**Abstract:** This research paper presents the application of the microwave infrared thermography (MIRT) technique for the purpose of detecting and characterizing defects in the carbon-fiber-reinforced polymer (CFRP) composite reinforcement of concrete specimens. Initially, a numerical model was constructed, which consisted of a broadband pyramidal horn antenna and the specimen. The present study investigated the application of a 360 W power system that operated at a frequency of 2.4 GHz, specifically focusing on two different operational modes: continuous and modulated. The specimen being examined consisted of a solid concrete slab that was coated with an adhesive layer, which was then overlaid with a layer of CFRP. Within the adhesive layer, at the interface between the concrete and CFRP, there was a defect in the form of an air gap. The study examined three distinct scenarios: a sample without any defects, a sample with a defect positioned at the center, and a sample with a defect positioned outside the center. The subsequent stage of the investigation incorporated experimental verification of the numerical modeling results. The experiment involved the utilization of two concrete specimens reinforced using CFRP, one without any defects and the other with a defect. Numerical modeling was used in this study to analyze the phenomenon of microwave heating in complex structures. The objective was to evaluate the selected antenna geometry and determine the optimal experimental configuration. Subsequently, these findings were experimentally validated. The observations conducted during the heating phase were particularly noteworthy, as they differed from previous studies that only performed observation of the sample after the heating phase. The results show that MIRT has the potential to be utilized as a method for identifying defects in concrete structures that are reinforced with CFRP.

**Keywords:** active thermography; microwave heating; fine-element method; defect; CFRP reinforcement

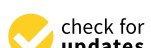



## 1. Introduction

### 1.1. Quality Control of the CFRP for Concrete Reinforcement

The utilization of carbon-fiber-reinforced polymer (CFRP) for the reinforcement of concrete is a progressively favored technique for the production of structures that exhibit enhanced resistance to structural damage [1–4]. The application of CFRP reinforcement typically involves the adhesion of an epoxy-resin-bonded layer composed of multiple carbon mats onto the cement substrate. The primary concern in this methodology pertains to the quality of the cement–CFRP adhesion [5,6]. The absence of adhesive bonding or the occurrence of delamination results in a localized reduction in the strength of the reinforcement, thereby creating a favorable environment for potential harm to the constructed edifices [7].

Consequently, it is of utmost significance to control said structures with regard to evaluating the quality of the bond between the concrete and composite. Various non-destructive testing techniques can be employed to monitor the quality of the cement–composite joint. These methods include radiographic, ultrasonic, and thermo-vision techniques [8–11]. In light of the dimensions of the structures under scrutiny, techniques that facilitate a comprehensive evaluation of vast regions are favored in this instance. In this aspect, active thermography was shown to be highly advantageous due to its ability to ensure prompt analysis of large structures and can be a non-contact method, which holds significant relevance in the domain of field research. The present study employed the active thermography technique with microwave excitation (MIRT).

*1.2. Active Thermography Method*

Active thermography involves the utilization of an external energy source to generate a thermal contrast within the examined specimen [12]. The investigation of thermal imaging techniques for potential application in the analysis of large-scale objects, such as complex concrete structures, is conducted in many research centers around the world. One potential approach involves the utilization of passive thermography, which is a technique that relies solely on natural sunlight, as opposed to active thermography, which utilizes an external heat source. It is important to take into account the fact that the implementation of this methodology requires extended periods of observation, with the outcomes dependent on factors such as diurnal variations and meteorological conditions [13]. Many techniques involve the utilization of active thermography, wherein conventional heat sources, such as halogen lamps, flash lamps, infrared radiators, and hot air [12–20], are employed. In this case, the results are no longer dependent on the external condition, but again, the main problem is the extended duration required for heating, which can be attributed to the substantial size of the structure under examination. In this case, it is important to bear in mind that the transfer of heat within the material takes place via conduction, wherein heat is transmitted from the heated surface to the interior of the object. The inherent specificity of these forms of excitation results in a substantial increase in inspection time. One potential approach to address this issue involves the utilization of vibrothermography techniques [21–23] (including classical, burst, and self-heating vibrothermography), which use dissipation processes and mechanical energy for defect detection. The excitation in this instance takes the shape of an acoustic wave, which passes through the tested structure and dissipates in damaged areas and defects. Although this is very effective, the acoustic methods typically involve the necessity of establishing direct contact between the exciter and the test object, which is a requirement that may present challenges in certain applications. Moreover, these methods are most appropriate for the identification of deeply located defects, such as cracks. The present study suggests the utilization of microwave irradiation as a heating method. Microwave heating has been widely employed in the literature for various purposes related to concrete structures, including expediting the curing process, decontaminating cement, and facilitating the drilling or melting of concrete [24–27]. In this proposal, microwaves are suggested as a potential energy source for infrared thermography [28–34]. The primary benefit of utilizing microwave heating lies in its volumetric nature. The rapidity of this technique is attributed to the heating of a specific volume of the specimen at a given time. Conversely, with regard to this alternative energy source, the heating ratio is contingent not solely upon the thermal characteristics of the substance but also upon certain electrical properties, especially the dielectric constant [24,35]. An interesting phenomenon is observed when microwaves interact with conductive materials, such as CFRP. The heating effect is constrained by a lower penetration depth value. Following the application of heat, a thermography imaging device is utilized to visually examine the surface of the specimen under inspection. Non-uniform heating of a sample can be attributed to the presence of inner material flaws, such as delamination and lack of adhesive. These flaws result in the manifestation of hotter or cooler regions visible in the temperature distribution on the sample's surface [36–40].

### *1.3. Novelty and Significance of the Research*

The present study aimed to examine the feasibility of utilizing active microwave thermography as a means of investigating reinforced concrete structures that were strengthened with carbon-fiber-reinforced polymer (CFRP). Conventional heating methods, such as halogen or flash lamps, or contact techniques utilizing hot pots, were suggested in earlier publications on this issue [41–43]. In our previous publication [44], where we suggested using the active microwave thermography approach, only a preliminary experimental study was conducted. In the work that is being presented here, we proposed a thorough study of the issue that took into account both numerical modeling and experimental investigation. First, a numerical model was developed to analyze the phenomenon being investigated. The objective of the model was to replicate the laboratory configuration and to examine scenarios that were not executed in the experiment, with the aim of assessing the applicability of the approach in a wider context. Consequently, we constructed models that depicted the sample's heating in both modulated and continuous modes. Furthermore, a defect-free sample and a centrally located defective sample were simulated, as per the experimental conditions. Additionally, a scenario where the defect was not centrally located was considered. It was shown that the modulated approach gave better results in terms of the time needed to obtain a detectable thermal signature of the defect. Moreover, the numerical results showed that the microwave heating was not uniform, and the higher temperature was obtained at the sample's center, which may affect the detectability of the defects. Analysis of the additional scenario—namely, a sample with the defect located outside the center—showed that it was possible to detect the defects in any location within the heated region of the sample. Subsequently, the findings obtained from numerical research were subjected to experimental validation. The conducted experimental tests—which were performed on two distinct samples, namely, one with a defect and one without, in the modulated regime—exhibited a significant level of concurrence with the numerical outcomes. It was shown that the constructed model possesses the capability to be employed for subsequent, complex investigations of the analyzed phenomenon.

### *1.4. Organization of the Paper*

The paper is structured as follows: The introduction discusses the problem of quality control in CFRP-reinforced concrete and introduces active thermography as a proposed method of non-destructive testing of such structures, with special emphasis on the microwave heating approach presented herein. Section 2 presents a numerical model along with illustrative outcomes for each scenario. In Section 3, the laboratory configuration is expounded upon, including the specimens utilized for experimentation and the outcomes obtained from the said experiment, as well as the findings according to the numerical outcomes. The fourth section of this article presents a discussion that provides a summary of its content.

### 2. Numerical Model

The numerical model presented in this manuscript was developed within the commercial software platform COMSOL. The finite element method (FEM) was employed to perform the calculations. The modeling of microwave infrared thermography (MIRT) necessitates the examination of two phenomena in a multi-physics mode, namely, electromagnetic wave propagation (EMW) and heat transfer (HT). The calculations were performed in the frequency (for EMW) and time domains (HT) (for continuous excitation) and in the time domain (for modulated excitation). The subsequent subsections present the specifics regarding the numerical modeling. First, the geometric characteristics of the model and the utilized mesh are demonstrated. Subsequently, the boundary conditions used are explicated, in conjunction with the material properties. Lastly, the outcomes of the model are illustrated and analyzed.

### 2.1. Model Geometry

The broadband horn antenna that was utilized had initially been designed using computer-aided design (CAD) software, specifically AutoCAD 2022. The dimensions of the antenna's aperture were 590 mm × 560 mm, while its height measured 660 mm. The antenna was connected to a rectangular waveguide that had dimensions of 36 mm × 62 mm and a length of 150 mm. Figure 1 illustrates the dimensions and design of the antenna under consideration. The aforementioned geometry was imported into the computational modeling software COMSOL. The geometry of the antenna enabled the stimulation of the specimen with a larger surface area, which is useful for the structures found in the civil engineering field. The dimensions of the aperture allow for heating samples with an almost 1 m width (with the inclination direction of the antenna). The antenna used in the present system had a gain of 21.67 dB in its main direction. The radiation pattern measurements were made in the far field by a sensor (placed at 25 m from the aperture) in the vertical and horizontal directions. The radiation pattern (H-plane and E-plane) of the antenna was considered in a polar axis system [29].

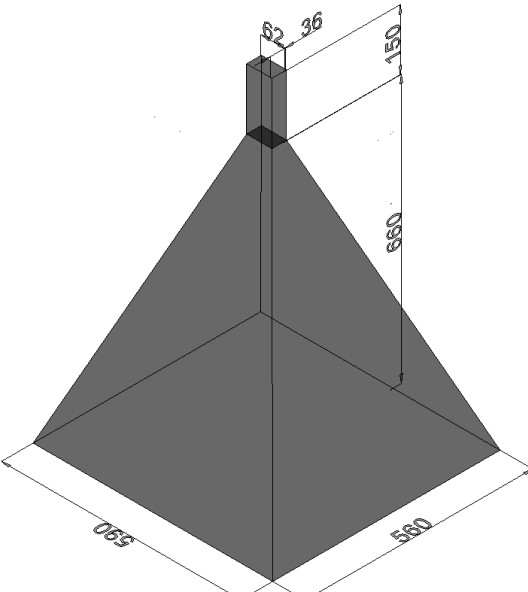

**Figure 1.** Geometry of utilized horn antenna with dimensions (dimensions in mm).

A block measuring 400 mm × 400 mm × 45 mm was utilized to model the test sample, which was a CFRP-reinforced concrete slab. The composite reinforcement was represented as a bilayer structure that comprised an epoxy adhesive layer measuring 1 mm in thickness and a layer of CFRP composite with a thickness of 1 mm. The defect in the form of a lack of adhesive was represented as a void space within the layer of epoxy adhesive. This type of defect can be caused by mechanical actions during the service life of civil engineering structures reinforced with CFRP. The dimensions of the defect were 100 mm × 100 mm and 1 mm in height. Three cases were simulated, namely, defect-free, centrally located defect, and off-center defect. Figure 2 displays all the dimensions and sample designs.

The specimen was positioned at a distance of 200 mm from the central point of the antenna aperture and was rotated at an angle of 45° with respect to the antenna. Figure 3a shows the full geometry of the model. In order to minimize the calculation time and the necessary computational resources, the simulation covered only 1/2 of the model presented in Figure 3a. The division of the model is illustrated in Figure 3b.

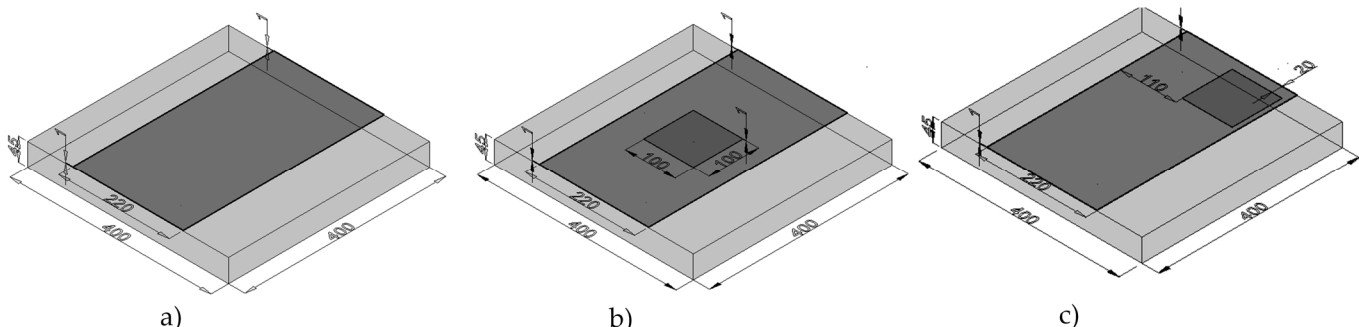

**Figure 2.** Models of the samples: (**a**) case without the defect, (**b**) case with the defect located in the center of the sample, and (**c**) case with the defect located away from the center of the sample (dimensions in mm).

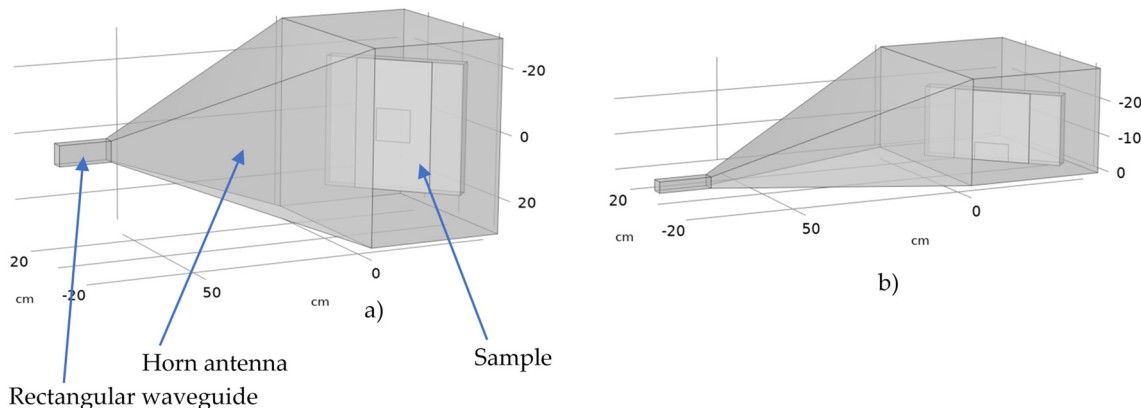

Horn antenna

Sample

Rectangular waveguide

**Figure 3.** Geometry of the model simulated in COMSOL: (**a**) the view on the full model and (**b**) half of the model used in the simulations.

### 2.2. Boundary Conditions and Material Properties and Meshing

As previously stated, the modeling of microwave heating necessitates a comprehensive examination of the issue through a combination of two distinct fields of physics: electromagnetic wave propagation and heat transfer. The model presented in this study described the propagation of an electromagnetic wave through the utilization of the electromagnetic wave equation, specifically in the frequency domain [45]:

$$\nabla^2 \mathbf{E} + \left( \mu \epsilon \omega^2 - i\mu\sigma\omega \right) \mathbf{E} = 0 \tag{1}$$

where $\mu$ denotes the magnetic permeability, $\epsilon$ is the electric permittivity, $\sigma$ represents the electric conductivity, $\omega$ is the angular frequency, and $\mathbf{E}$ is the electric field vector. This equation is solved in all domains within the model. Conversely, the heat transfer was characterized by the heat balance equation defined as follows [12]:

$$\rho C_p \left( \frac{\partial \mathrm{T}}{\partial t} + \nabla \mathrm{T} \right) + \nabla \mathbf{q} = Q \tag{2}$$

where $\rho$ is the material density, $C_p$ denotes the material heat capacity, $\mathbf{q}$ is the heat flux associated with the convection phenomenon, T is the temperature, and Q denotes the external heat source. This phenomenon was exclusively taken into account within domains that were associated with the sample. The external heat source was equal to the resistive heat generated by the electromagnetic field, which is described as follows (derived from Poynting's theorem [45]):

$$Q = 1/2\mathrm{Re}((\sigma - j\omega\epsilon)\mathbf{E}\cdot\mathbf{E}^*) \tag{3}$$

where Re indicates the real part of the value. Accurately simulating the phenomenon under investigation necessitates the identification of suitable boundary conditions for both electromagnetic waves and heat transfer. The simulated antenna's walls were subjected to an impedance boundary condition, which is defined as follows [46]:

$$\sqrt{\frac{\mu}{\epsilon - j\sigma/\omega}}\mathbf{n} \times \mathbf{H} = \mathbf{n} \times \mathbf{E} \tag{4}$$

where **H** is the magnetic field strength vector and **n**. is the normal vector. The walls of the air-filled domain surrounding the sample were subjected to a scattering boundary condition [47]:

$$\mathbf{n} \times (\nabla \times \mathbf{E}) - jk\mathbf{n} \times (\mathbf{E} \times \mathbf{n}) = 0 \tag{5}$$

where $k$ is the wavenumber. An electromagnetic wave was produced through the application of the port boundary condition on a specific wall within the waveguide. Here, the assumed mode was $TE_{10}$ and the propagation constant was equal to [48]

$$\beta = \frac{\omega}{c}\sqrt{1 - \frac{\omega_c^2}{\omega^2}}, \tag{6}$$

where $\omega_c = c\sqrt{\left(\frac{n\omega}{a}\right)^2 - \left(\frac{m\omega}{b}\right)^2}$ is the cut-off frequency of a waveguide, where $n = 1$ and $m = 0$. Finally:

$$\beta = \frac{2\pi}{c}\sqrt{f^2 - \frac{c^2}{4a^2}}, \tag{7}$$

where $a = 62$ mm was the width of the rectangular waveguide, $c$ denotes the speed of light, and $f$ denotes the frequency. The perfect magnetic conductor condition was imposed on the plane of symmetry, which served to define the division of the model:

$$\mathbf{n} \times \mathbf{H} = 0 \tag{8}$$

Regarding the HT domain, which was confined solely within the sample, the convective flux condition was enforced on all walls of this domain as the boundary condition [12]:

$$q_c = h(T_{ext} - T) \tag{9}$$

where $h$ denotes the heat transfer coefficient and $T_{ext}$ is the external temperature value. The boundaries with specified boundary conditions are presented in Figure 4.

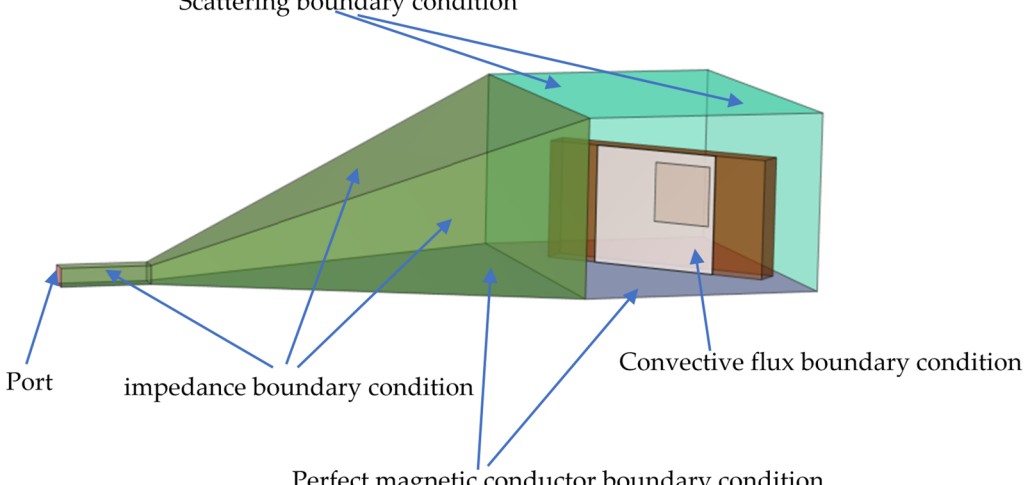

**Figure 4.** Specification of the boundary conditions used in the modeling process.

The model was computed in two modes: continuous (with an equal power of 360 W applied for all the time of simulation) and modulated with the power applied with a function $f(t)$ presented in Figure 5:

$$P_{mod} = 360[\text{W}] \cdot f(t) \tag{10}$$

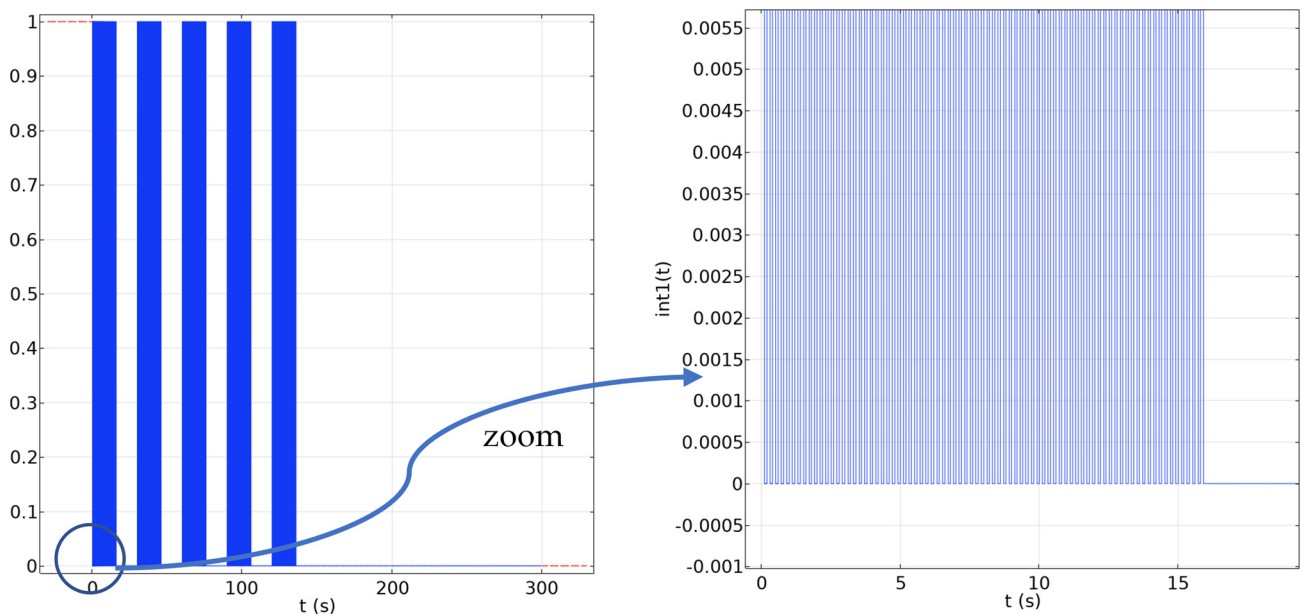

**Figure 5.** Diagram of the function used for simulating the modulated excitation.

For the continuous mode, the study consisted of two steps: a frequency domain computation step (solved only for EMW) followed by a time-dependent computation step (only for HT). In this mode, the power was assumed to be constant. For the modulated mode, the time-dependent step was solved for both EMW and HT.

In the presented simulation, the microwave antenna was modeled as an air-filled structure with aluminum edges. The structure of the sample was described previously. Table 1 presents the selected properties of the materials used in the simulation.

**Table 1.** The material properties utilized in the modeling [49,50].

|  | Air | Aluminum | Epoxy | CFRP | Concrete |
|---|---|---|---|---|---|
| Cp (J/(kg·K)) |  | N/A | 1700 | 1700 | 750 |
| Density (kg/m³) |  | N/A | 1150 | 1150 | 2400 |
| Epsilon (relative permittivity) | 1 | 1 | 3.9–2.2 j | 3.6 | 8.4–2.86 j |
| Thermal conductivity (W/(m·K)) |  | N/A | 0.18 | 5 | 0.8 |
| Electrical conductivity (S/m) | 0 | $3.77 \cdot 10^7$ | 0 | 0.5 | 0 |
| Relative permeability | 1 | 1 | 1 | 1 | 1 |

The number of elements in the simulated model was dependent on the analyzed case. The utilized mesh consisted of 76,093 tetrahedral elements for the case with no defect, 79,677 elements for the case with a defect located in the center of the sample, and 80,049 for the case with a defect located outside the center. The generated mesh is shown in Figure 6. As can be noticed, the sizes of the elements changed between the domains. The mesh refinement process was performed manually by utilizing the scale option provided in the COMSOL software. This approach was specifically applied to the domains that encompassed the sample, as these regions were associated with calculations related to both the utilized physics: EMW and HT.

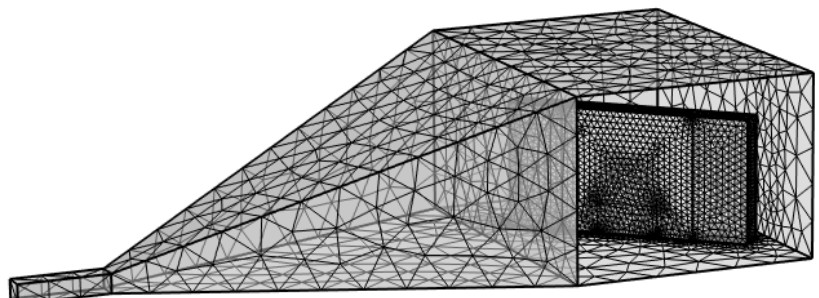

**Figure 6.** Utilized mesh.

## 2.3. Exemplary Results

Selected results of numerical simulations are presented below. In the first stage, in order to better illustrate our results, the temperature distribution on the surface of the sample is presented in a system that depicts the entire simulated area (including the antenna). Figures 7 and 8 show the temperature distributions represented on the surface of the specimen, which solely encompassed CFRP concrete reinforcement, for two distinct forms of excitation: continuous (Figure 7) and modulated (Figure 8). The presented results depict the temperature distribution for three cases, namely, without the defect (Figures 7a and 8a), the defect located at the center of the sample (Figures 7c and 8c), and the defect located outside the center of the sample (Figures 7b and 8b). These results were obtained after a heating duration of 100 s. It can be easily seen that in all cases, the sample was heated unevenly, with the maximum temperature around its center (Figures 7a and 8a). This, of course, was related to the specific radiation pattern of the selected antenna. However, it was possible to distinguish between a non-defective and a defective sample due to the fact that the area around the defect was heated to a greater extent. We present a more detailed analysis below.

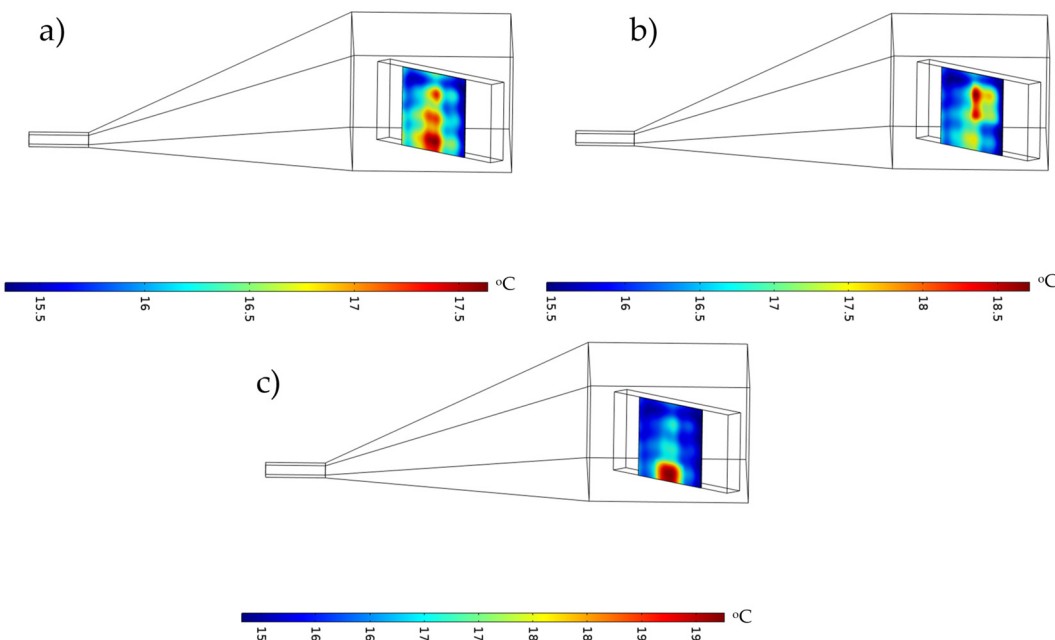

**Figure 7.** The raw results of the numerical modeling for the case of continuous excitation presented for the chosen time step (t = 100 s): (**a**) the case without the defect, (**b**) the case with the defect located away from the sample's center, and (**c**) the defect located in the center of the sample.

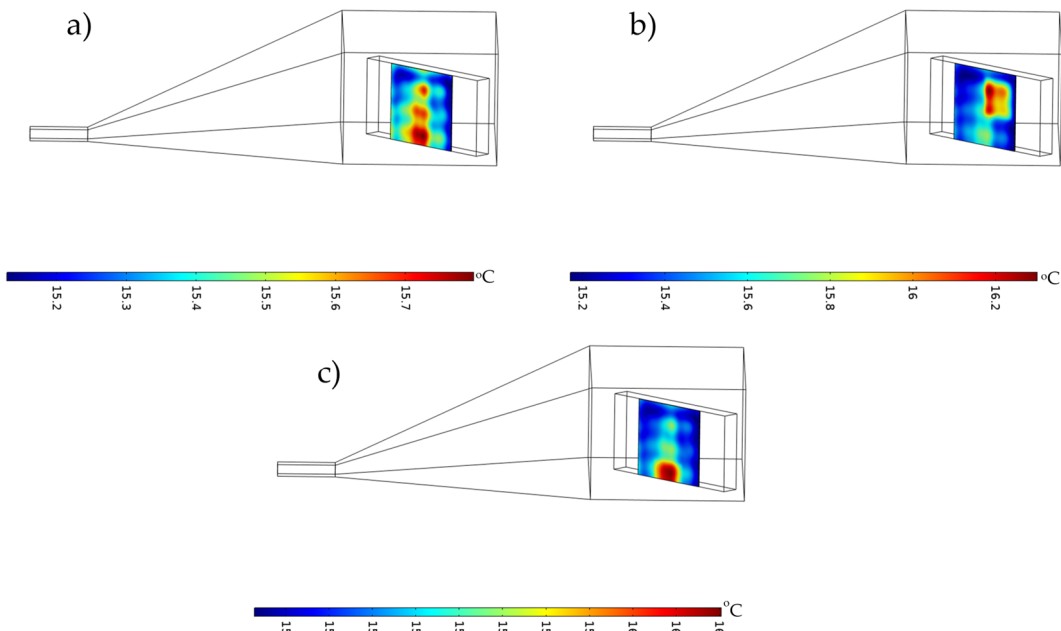

**Figure 8.** The raw results of the numerical modeling for the case of modulated excitation presented for the chosen time step (t = 100 s): (**a**) the case without the defect, (**b**) the case with the defect located away from the sample's center, and (**c**) the defect located in the center of the sample.

The following section provides a broader overview of the results. The presented Figures 9 and 10 depict thermograms that were selectively chosen for both continuous and modulated excitation, respectively. The complete thermogram of the sample, both without a defect and with a centrally located defect, was reconstructed through mirror reflection (Figure 9a,b and Figure 10a,b). It should be noted that in the case of a system featuring an off-center defect, the aforementioned operation was not feasible. Consequently, in this case, we solely present one-half of the surface, as it was simulated in COMSOL (Figures 9c and 10c). The results of the study are shown in the form of unprocessed temperature distributions, namely, thermograms, in which the temperature values were standardized to a singular temperature scale. The outcomes for three specific time intervals are demonstrated, namely, at 50 s, 100 s, and 150 s, which corresponded to the end of the heating process. The data indicate that the temperature under continuous excitation was notably higher compared with that under modulated excitation, with maximum values of 18.88 °C and 16.11 °C, respectively. However, it is noteworthy that in both instances, the defect became apparent at the 50 s mark and the qualitative outcomes were similar. For instances of continuous heating, the central region of the sample experienced a considerably higher degree of heating compared with its off-center counterpart. This is a crucial factor to consider when the defect is located away from the center. The enhanced detectability of the off-center defect under modulated excitation was readily apparent.

Below, the time–temperature characteristics illustrating the temporal fluctuations in the mean temperature within a specified region are shown. For this purpose, specific domains of interest were chosen for each case. For samples without the defect or a central defect, the designated region of interest (ROI) was the area in the middle that encompassed the defect. When dealing with a specimen that presented an off-center flaw, the ROI corresponded to the specific area that covered the defect. Figure 11 depicts specific regions for each case.

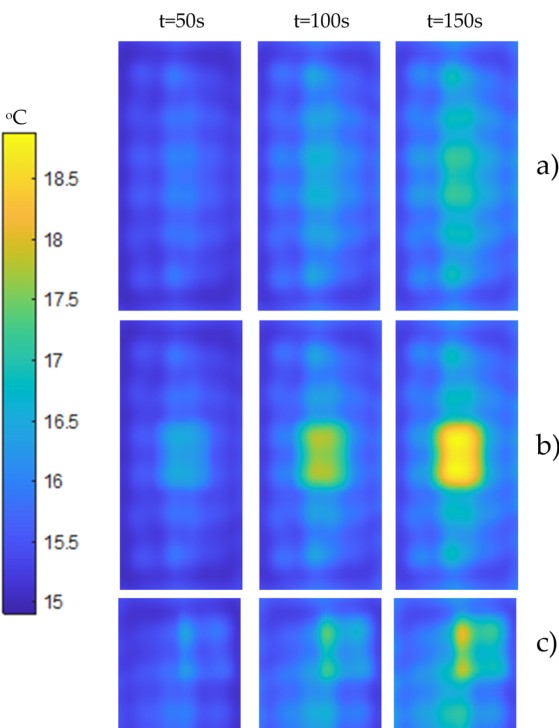

**Figure 9.** The results of the numerical modeling for the case of continuous excitation presented for the chosen time steps (50 s, 100 s, and 150 s): (**a**) the case without the defect, (**b**) the defect located in the center of the sample, and (**c**) the case with the defect located away from the sample's center. The thermograms in (**a**,**b**) were reconstructed using mirror reflection.

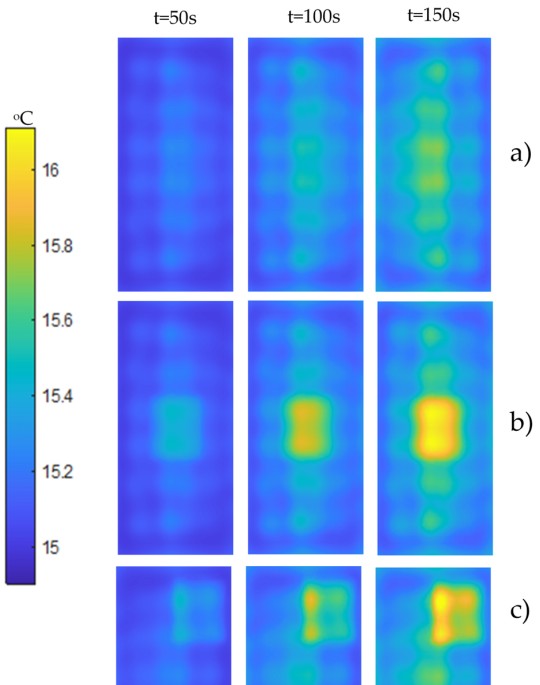

**Figure 10.** The results of the numerical modeling for the case of modulated excitation presented for the chosen time steps (50 s, 100 s, and 150 s): (**a**) the case without the defect, (**b**) the defect located in the center of the sample, and (**c**) the case with the defect located away from the sample's center. The thermograms in (**a**,**b**) were reconstructed using mirror reflection.

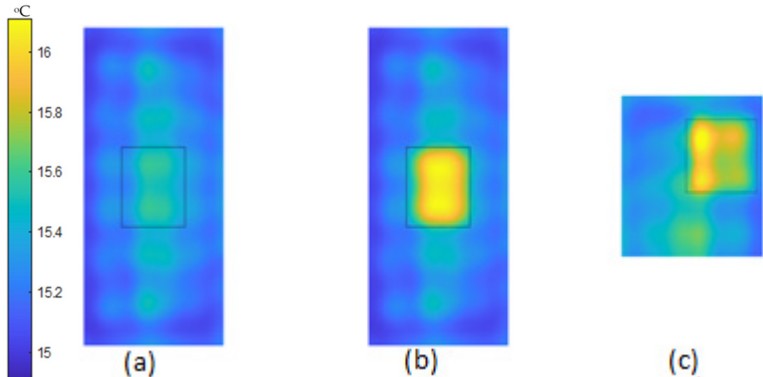

**Figure 11.** Chosen ROIs for all the simulated cases: (**a**) without the defect, (**b**) defect located centrally, and (**c**) defect away from the sample's center.

The data presented below displays the time–temperature characteristics for all cases for both continuous and modulated excitation (Figure 12). Evidently, in the case of continuous excitation, there was a marked escalation in the overall temperature, particularly in relation to the centrally positioned defect. The disparity between the peak of the mean temperature for the region exhibiting the defect (as represented by the blue solid line in Figure 12a) and that of the region devoid of the defect (as depicted by the red dotted line in Figure 12a) was equal to 1.78 °C. In the context of comparison, it can be observed that for modulated excitation, the aforementioned difference amounted to 0.31 °C. It is noteworthy that these variations can be detected through the utilization of highly sensitive thermal imaging cameras. The sample with a defect located outside its central region presented a more intriguing scenario. In regard to both excitations, there existed a disparity of approximately 0.28 °C between the peak of the mean temperature for the region with a defect (indicated by the yellow dashed line in both Figure 12a,b) and the region without a defect (indicated by the red dotted line in both Figure 12a,b). In the scenario where the excitation was modulated, the observed distinction bore a striking resemblance to the situation involving a central defect. This implies that this approach may be more efficacious in instances where numerous defects are distributed throughout the sample in diverse locations. Significantly, in the context of modulated excitation, the temperature differential between the regions with and without the defect became evident within approximately 15 s of heating. In contrast, under continuous excitation, this temperature differential manifested after approximately 35 s. This likely implies enhanced operational speed of the modulated excitation-based approach.

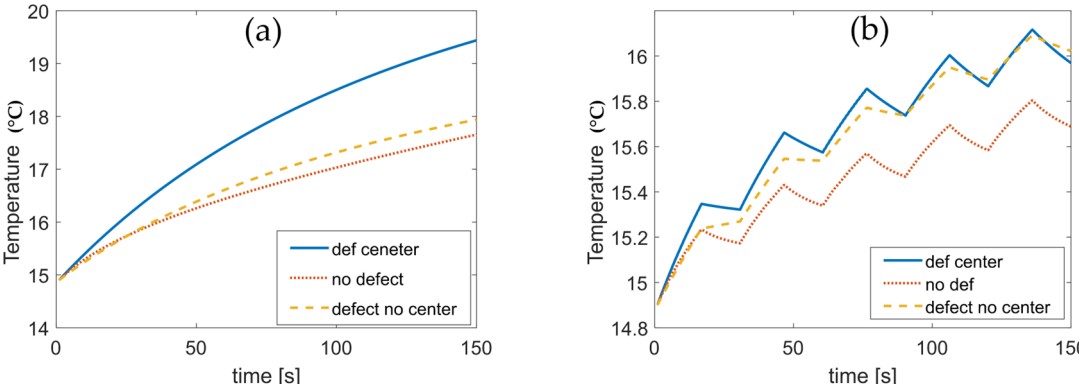

**Figure 12.** Average temperature dynamics computed from the chosen ROI: (**a**) for continuous excitation and (**b**) for modulated excitation. In the legend, "def center" denotes the sample with the defect located in the center, "no defect" indicates the sample without the defect, and "defect no center" is the sample with the defect located outside the center.

## 3. Experimental Tests and Results

### 3.1. Samples and Experimental Setup

Two experimental campaigns were carried out: one with a CFRP sample without a defect and another test with a sample with a defect, which corresponded to two cases of the numerical simulations (defect-free and a centrally located defect, respectively). The adhesive used in the study was Araldite® 2015, which is a two-component epoxy paste adhesive. It is particularly suitable for SMC and GRP bonding. The carbon fibers are the SikaWrap®-301 C. The composite laminate was defined as the stacking of two unidirectional plies with two different fiber orientations (perpendicular). The sample is depicted in Figure 13 shown below. The defect was represented by the presence of air (the absence of an epoxy adhesive layer with a thickness of 1 mm) over an area of 100 mm × 100 mm.

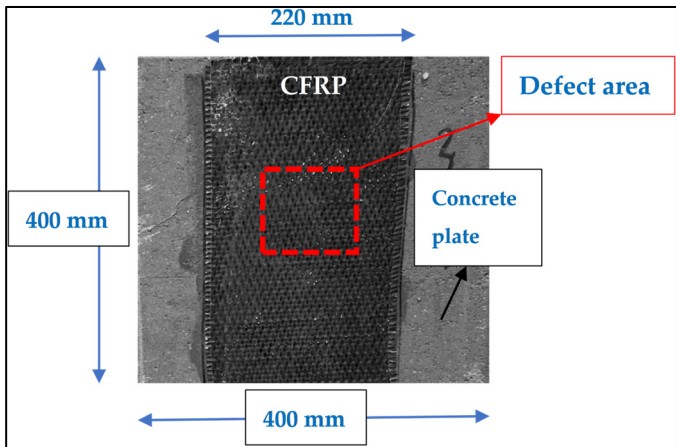

**Figure 13.** Examined sample with a visible layer of CFRP at the concrete surface. Red rectangle indicates the defect's position under the composite layer.

The experimental setup of the tests with the CFRP samples is shown in Figure 14. The same microwave system as in the numerical model, with a frequency of 2.45 GHz, was used to heat the samples. The pyramidal horn antenna was placed at the same position and in the same direction as in the numerical model (200 mm from the sample and with a 45° direction). A heating power of 360 W was used for heating the samples for 150 s. In order to obtain the surface temperatures of the heated specimen for comparison with the numerical model, a medium wave range (3 to 5 µm wavelength) infrared camera was used to record thermograms with a frame rate of 1 image per second. In order to detect the whole surface of the specimen, the following position of the infrared camera was chosen: 55° direction with a distance of 1.5 m from the sample. The thermograms were recorded for a duration of 475 s in order to observe the cooling phase of the experiments.

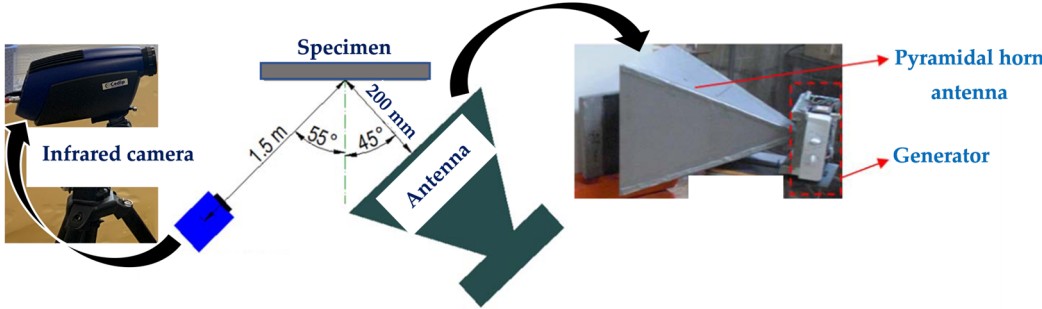

**Figure 14.** Experimental test setup with the CFRP specimen and MIRT.

### 3.2. Results from Experimental Campaigns

To compare with the results from the numerical modeling, thermograms obtained at three instants (50 s, 100 s, and 150 s) during the experimental tests with the sample without a defect and that with a defect are shown in Figure 15. In accordance with the numerical modeling (thermograms in Figure 10), the thermograms show the controlled defect area, which was hotter than the healthy area, on the specimen with a defect (Figure 15b). These elevated temperatures can be attributed to the presence of air, which showed a lower thermal conductivity compared with that of the epoxy adhesive. Another observation referred to the impact of the non-uniformity of the microwave beam on the thermograms of the defect-free specimen, resulting in the generation of additional areas of increased temperature (Figure 15a). That was also the case in the numerical modeling (Figure 10a).

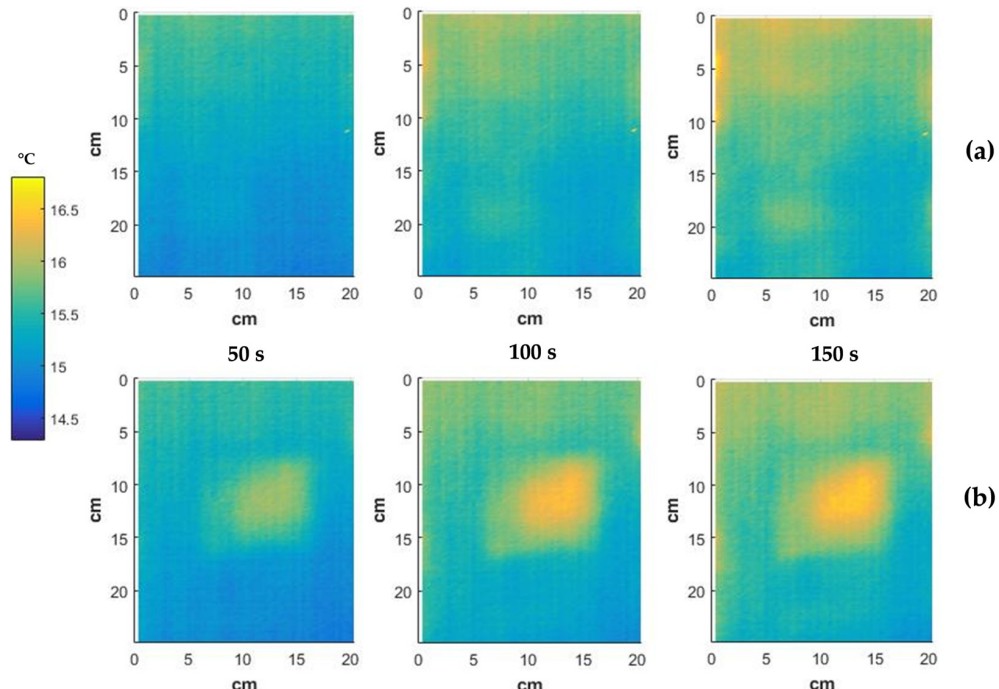

**Figure 15.** Thermograms at different instants: (**a**) sample without the controlled defect and (**b**) sample with the controlled defect.

The temporal evolutions of the temperature on the defect and non-defect regions for both samples are shown in Figure 16.

In accordance with the results from the numerical modeling (Figure 12b), the temperature curves clearly show modulation due to the excitation (microwave signal). After the heating phase (150 s), the modulation disappeared from the temperature curves. In the defect area (red solid curve), the temperature increased from 15.2 °C to 16.2 °C, while that of the defect-free region (blue dotted curve) increased only from 14.9 °C to 15.3 °C (Figure 16b). On the other hand, there was only a small difference between both observed regions on the sample without a defect (both curves in Figure 16a), although there was an effect of the non-uniformity of the microwave beam.

The temporal thermal contrast between both samples is also shown in Figure 17. Both samples may have had different initial surface temperatures before the test. This is why the initial temperatures were subtracted from the thermal contrast. Obviously, the maximum thermal contrast (0.7 °C) occurred at the end of the heating phase. It should be noted that it is the thermal contrast that allows us to see the defect area on the thermograms.

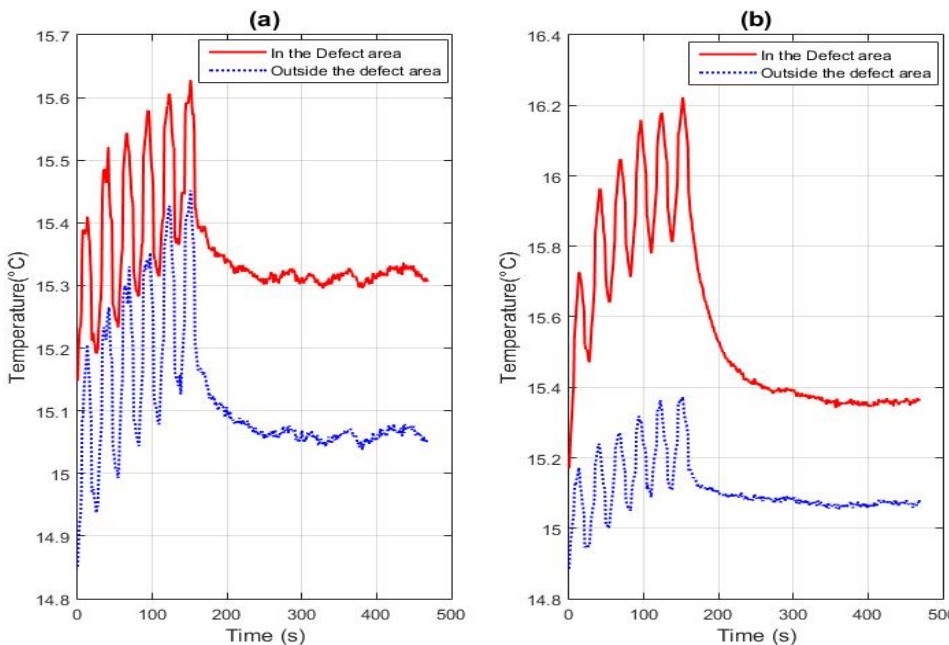

**Figure 16.** Time–temperature characteristics obtained for both samples: (**a**) sample without a controlled defect and (**b**) sample with a controlled defect.

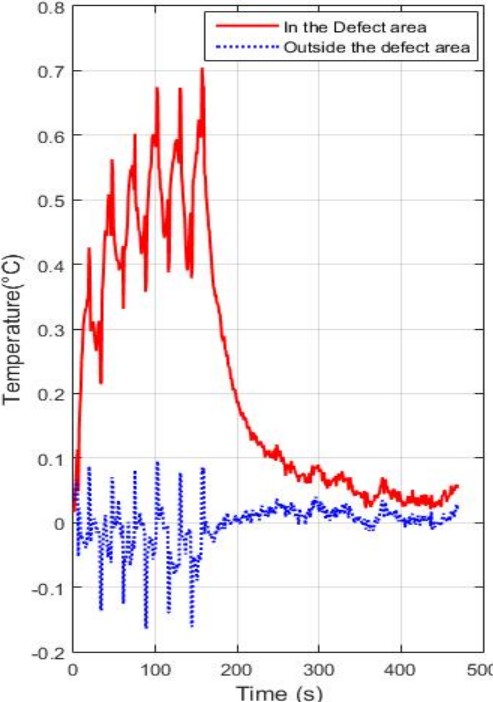

**Figure 17.** Calculated temperature difference, which was obtained by subtracting the initial temperature distribution from subsequent thermograms, for both defective and non-defective areas.

It is important to notice that the comparison between the experimental approach and the numerical modeling, especially the results with the modulated excitation, showed a higher temperature at the defect area than the area without a defect (due to the presence of the air having smaller thermal conductivity than that of the adhesive). Another common point of the results from both approaches is the influence of the non-uniformity of the microwave beam on the thermograms. This non-uniformity was generated by the reflection

of the waves with the antenna and the sample. This phenomenon must be considered in the detection with the MIRT method.

## 4. Conclusions

In the presented study, the feasibility of MIRT for monitoring reinforced concrete structures strengthened with CFRP was examined. For this purpose, a numerical model (with continuous and modulated excitation), corresponding to the laboratory setup in the same manner, was developed to accurately analyze the examined phenomenon and to apply scenarios that were not executed experimentally. The compliance of the results from both approaches (numerical and experimental) in the case of modulated excitation allowed for validating the numerical model for detecting a defect in a CFRP reinforcement. The defect area, which was represented by the absence of an adhesive (or the presence of the air), had a higher surface temperature than the temperature of the area without a defect. Both approaches yielded similar results, which emphasized the non-uniformity of the microwave beam. This phenomenon must be taken into account, as it can potentially lead to the misinterpretation of thermograms when attempting to identify defects. In addition to validating the numerical model via experimental testing, the broader context of the study was provided by the numerical model itself. This included determining the influence of the defect's position on its detectability and evaluating the efficiency of the modulated excitation method in comparison with continuous excitation. This study demonstrated the superiority of MIRT over alternative active thermography techniques, particularly in terms of its ability to ensure efficient and fast inspection, even for large structures. Furthermore, the non-contact nature of this method is of utmost significance in numerous instances of field research. Our previous research on the MIRT method demonstrated its versatility, making it applicable to various materials, particularly those with low conductivity (but not only these materials, as it can be employed to analyze surface defects in metals). The method's broad range of applications stems from its ability to detect both inner and shallow defects through volumetric heating. Active thermography is a technique that enables the detection of various types of defects, such as cracks, delamination, corrosion, voids, lack of adhesive, and inclusions. This method is particularly effective, as it exhibits sensitivity to any structural alterations present in the objects being tested.

The present study offers potential perspectives in terms of applying the MIRT method to investigate the durability of composites under various conditions, such as different types of defects or varying environmental conditions. Additionally, the research encompassed the advancement of the detection technique utilizing various excitation systems, such as alternative microwave generators and antennas. This was done to accommodate diverse types of defects and commonly employed composite materials. Our research team intends to proceed with the subsequent phases of the research outlined in this study. These phases include the analysis of specific structures to identify additional types of defects, such as moisture or material deterioration caused by reinforcement corrosion. Furthermore, we plan to explore the application of microwave systems with alternative configurations, specifically investigating the influence of electromagnetic wave frequency and frequency modulation on the outcomes obtained. Additionally, we aim to employ machine learning techniques, particularly artificial neural networks, to automate the assessment of identified defects.

**Author Contributions:** Conceptualization, S.A.K. and B.S.; methodology, S.A.K. and B.S.; formal analysis, S.A.K. and B.S.; investigation, S.A.K., C.L.R. and B.S.; data curation, S.A.K. and B.S.; writing—original draft preparation, S.A.K. and B.S.; writing—review and editing, S.A.K. and B.S.; visualization, S.A.K. and B.S.; supervision, D.D. and F.B.; funding acquisition, D.D. and F.B. All authors have read and agreed to the published version of the manuscript.

**Funding:** This research was partially funded by the National Science Center, Poland (Narodowe Centrum Nauki, NCN), within the research project "Evaluation of the internal structure and assessment of the structure health of complex materials using active infrared thermography with multiple excitation sources", grant number 2020/04/X/ST7/01388.

**Institutional Review Board Statement:** Not applicable.

**Informed Consent Statement:** Not applicable.

**Data Availability Statement:** Not applicable.

**Acknowledgments:** The work presented in the paper is the outcome of the collaboration between a research team in Poland (Active Thermography Laboratory) and two other research teams in France (LGCgE: Laboratoire de Génie Civil et géoEnvironnement, and ENDSUM of Cerema Angers: Evaluation Non-Destructive des StrUctures et des Matériaux).

**Conflicts of Interest:** The authors declare no conflict of interest.

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
