# Peer review of "Defect Detection in CFRP Concrete Reinforcement Using the Microwave Infrared Thermography (MIRT) Method—A Numerical Modeling and Experimental Approach"

_applsci, doi:10.3390/app13148393_

Round 1

Reviewer 1 Report

1- The abstract needs more interest and rewriting some paragraphs.

2- There are still some aspects that can be improved (for grammar and punctuations). Improve the technical writing of your paper, where there are several grammatical errors and spelling I think they need to be checked out.

3- The conclusion needs more efforts to elaborate the achieved results with respect to the future work,

4- The practical part is very important, therefor, I’m asking about the radiation pattern measurements,

5- Why this design geometry, explain the design methodology and specify the novelty with your antenna structure,

5- Future work is an important part of the conclusion.

6- There is several relative works, I wish to discuss, based on adding such structures in the patch such as following:

A-    Arkan Mousa Majeed, Taha A. Elwi, Zaid A. Abdul Hassain " Defected Slots 3D Antenna Structure for Millimeter Applications " , International Journal of Industrial Electronics and Electrical Engineering (IJIEEE) , Volume-10,Issue-4  ( Apr, 2022 ),

B-    M. A. Jawad, M. A. Elwi, E. Y. Salih, Taha A. Elwi, and Zulkifly Abbas, “Monitoring the Dielectric Properties and Propagation Conditions of Mortar for Modern Wireless Mobile Networks,” Progress in Electromagnetic Research Letters, Volume 89, pp. 91-97, January 2020.

C-    76.       Taha A. Elwi and Y. Alnaiemy, “Electromagnetic Characterizations of Cement Using Free Space Technique for the Application of Buried Object Detection”, Diyala Journal for Pure Science, volume 11, issue 4, pp. 1-10, July 2015.

I loved this work and I feel it is very good. I hope these comments would help you improve this work after a major revision.

Regards

In general, the English is good but needs further enhancement to make sentences much clear

Reviewer 2 Report

The authors have presented an interesting study on infrared tomography by microwave excitation of carbon fiber reinforced polymer (CFRP) composites. The experimental results are compared with simulations performed using COMSOL. The technique and setup described in this work have the potential for numerous characterizations. However, there are some minor revisions that the authors should consider before publication to further improve the paper.

1.       It would be beneficial for the authors to mention the possible effects of loading and orientation of carbon fibers as they pertain to the focus variable of the experiment and simulation, which is the defect.

2.       While the references are adequately cited in the introduction section, they seem to be lacking in the discussion section. To emphasize the contribution of their work to the existing knowledge, the authors should discuss their results in reference to recently published articles.

3.       Fig. 11 would benefit from the inclusion of a colored scale for better interpretation. Additionally, the caption of Fig. 13 requires additional information to enhance clarity.

4.       Please avoid excessive use of the phrase 'It is worthy to remind that.'

5.       The authors should carefully proofread the manuscript to eliminate typographic errors and ensure consistency. Here are a few examples:

•        The acronym 'MIRT' should be removed from the title as it stands for Microwave InfraRed Thermography, not microwave thermography method.

•        The full term 'Carbon Fiber Reinforced Polymer' should not be repeated in the conclusion section since the abbreviation 'CFRP' has been previously used.

•        The numeral '3' in the first column of Table 1 should be in superscript.

•        In Line 185, one full stop should be removed for proper punctuation.

Addressing these revisions will enhance the overall quality of the paper and improve its clarity and readability.

Moderate editing of English language required

Reviewer 3 Report

manuscript ID: applsci-2490600-peer-review-v1

Despite the efforts done by the authors, and the manuscript contains a good story, there are some important points that should be considered by the authors.

1.      Abstract is long, it should include a sentence introduction, the purpose of the work, and the main findings

2.      What is the novelty of the current work? It should be clarified at the end of the introduction.

3.      In caption of Figure 1, please change (dimensions in [mm]) to (dimensions in mm). delete square brackets.

4.      In Eq.1, what do you mean by i.  Re in Eq.3 All symbols shoud be clarified in all equations. 

5.      Please add the missing references to the equations deduced by other authors.

6.      What do you mean by GRFP in Table 1.

7.      In legend Fig.12. Please use the same expression. What do you mean by def center, no def and defect no center .

8.      Please write a complete caption  for Figure 13 instead of Figure 13. Sample!!!!!.

9.      Caption of Figure 16. Temperature evolution….  "I think Temperature evolution against time". Please check also caption of Fig.17,. Figure 17. Temperature increase.. with what ???

10.  What is the next step for the current study? Future work.

Reviewer 4 Report

In this manuscript, the authors presented the results of studies on the defect detection in CFRP reinforcement structures in civil engineering using microwave thermography technique supported by numerical modeling. The Introduction is generally well written and problem is properly formulated, some minor suggestions regarding this section were given in the detailed comments. Next, the authors described the physics of the investigated technique and the results of simulations. Some extensions regarding FE model are necessary. In section 3, the authors presented the experimental setup and results obtained from experiments. The comparison of numerical and experimental results was performed. Revisions of the manuscript are necessary.

1.     The abstract is too long and contain too much details, like specific dimensions of specimens and details on experimental setup. It is recommended to shorten the abstract and rewrite it in a more general way.

2.     Please add explanation/justification of selection of IRT from the described NDT techniques in the first paragraph of the Introduction.

3.     It is suggested to extend the review of IRT techniques, besides the mentioned classical ones, there are techniques that belong to the class of vibrothermography, like classical and burst vibrothermography, self-heating vibrothermography. Such an overview will help strengthening the justification of selection of MIRT.

4.     Please provide the source for data presented in Table 1.

5.     Please provide more details on the FE model: type of elements, way of meshing, boundary conditions, solvers, etc.

6.     Please provide detailed information on specimens: manufacturer, stacking sequences, type of reinforcement, etc.

7.     It is essential to discuss the advantages of MIRT with respect to other IRT techniques to show its superiority also in practical applications.

Round 2

Reviewer 3 Report

I accept the changes made and explanations sent 

Reviewer 4 Report

The revised version of the manuscript was significantly improved. The authors answered all comments and performed necessary explanations and extensions in the manuscript. Please check the correctness of names and surnames of the authors in the added references. After these corrections, the manuscript can be considered for a publication in its present form.